# Reversible conversion between skyrmions and skyrmioniums

Sheng Yang [1,10], Yuelei Zhao[1,10], Kai Wu[1], Zhiqin Chu [2,3], Xiaohong Xu[4,5], Xiaoguang Li [6] ✉, Johan Åkerman [7,8,9] ✉ & Yan Zhou [1] ✉

Skyrmions and skyrmioniums are topologically non-trivial spin textures found in chiral magnetic systems. Understanding the dynamics of these particle-like excitations is crucial for leveraging their diverse functionalities in spintronic devices. This study investigates the dynamics and evolution of chiral spin textures in $[Pt/Co]_3/Ru/[Co/Pt]_3$ multilayers with ferromagnetic interlayer exchange coupling. By precisely controlling the excitation and relaxation processes through combined magnetic field and electric current manipulation, reversible conversion between skyrmions and skyrmioniums is achieved. Additionally, we observe the topological conversion from a skyrmionium to a skyrmion, characterized by the sudden emergence of the skyrmion Hall effect. The experimental realization of reversible conversion between distinct magnetic topological spin textures represents a significant development that promises to expedite the advancement of the next generation of spintronic devices.

Magnetic topological textures have attracted intensive attentions in recent years due to their unique properties in the field of emergent electrodynamics[1–7]. In particular, fast skyrmion motion can be induced by an applied electric current[5,8–10], which could potentially be used in practical applications such as data storage and logic computing devices[1,3,11]. However, a ferromagnetic skyrmion driven by a current is subject to the skyrmion Hall effect (SkHE)[12–17], in which the skyrmion also moves in a transverse direction to that of the current. This effect has hindered the use of skyrmions in practical applications. Researchers have offered several potential solutions, including synthetic antiferromagnet skyrmions[18–20], compensated ferrimagnetic skyrmions[21–24], and skyrmioniums[25–34]. Skyrmioniums always have a topological number $Q = 0$, and have been recently investigated both

theoretically and experimentally in a frustrated Kagome magnet $Fe_3Sn_2$[32], and a FeGe bulk device[28]. Tang et al. have verified that the SkHE can be suppressed to zero in the Bloch-type skyrmionium[35]. The Néel-type skyrmionium is also regarded as a superior information carrier compared to the conventional skyrmion, mainly due to its advantages in current-driven dynamics[34]. Unfortunately, the experimental creation, manipulation, and conversion of a Néel-type skyrmionium from a regular skyrmion using current still pose significant challenges.

In the past decade, conversions between skyrmions and other magnetic textures, such as domain walls[36] and merons[37], have been studied. The conversion between skyrmions and skyrmioniums is a more complicated and challenging process, involving multiple

[1]School of Science and Engineering, The Chinese University of Hong Kong, Shenzhen 518172, China. [2]Department of Electrical and Electronic Engineering, The University of Hong Kong, Hong Kong 999077, China. [3]School of Biomedical Sciences, The University of Hong Kong, Hong Kong 999077, China. [4]Research Institute of Materials Science of Shanxi Normal University & Collaborative Innovation Center for Shanxi Advanced Permanent Magnetic Materials and Technology, Linfen 041004, China. [5]School of Chemistry and Materials Science of Shanxi Normal University & Key Laboratory of Magnetic Molecules and Magnetic Information Materials of Ministry of Education, Linfen 041004, China. [6]Shenzhen Key Laboratory of Ultraintense Laser and Advanced Material Technology, Center for Advanced Material Diagnostic Technology, and College of Engineering Physics, Shenzhen Technology University, Shenzhen 518118, China. [7]Department of Physics, University of Gothenburg, Gothenburg 41296, Sweden. [8]Center for Science and Innovation in Spintronics, Tohoku University, 2-1-1 Katahira, Aoba-ku, Sendai 980-8577, Japan. [9]Research Institute of Electrical Communication, Tohoku University, 2-1-1 Katahira, Aoba-ku, Sendai 980-8577, Japan. [10]These authors contributed equally: Sheng Yang, Yuelei Zhao. ✉e-mail: lixiaoguang@sztu.edu.cn; johan.akerman@physics.gu.se; zhouyan@cuhk.edu.cn

conversions of different magnetic structures. Although some theoretical reports have described single-step conversions[25,27,29,30,34], the experimental realization of reversible conversion between these two topological spin textures requires precise control and manipulation of the domain wall through applied magnetic fields or electric currents. In this study, we experimentally realize reversible conversion between skyrmions and skyrmioniums at room temperature in a ferromagnetic multilayer system. We first observe that a skyrmion can be expanded to a high-energy transition state with sinusoidal current pulses at a perpendicular field of 6 mT, which can be transformed into a skyrmion bag[38] and further a skyrmionium by deleting the stripe domains with sinusoidal current pulses at a perpendicular field of 10 mT. The skyrmion bag can also be driven by square pulses and then converted to a skyrmionium. We also directly observe the topological conversion from a skyrmionium to a skyrmion, characterized by the sudden emergence of the skyrmion Hall effect. The experimentally observed morphology and dynamics of the spin textures are corroborated by micromagnetic simulations.

## Results

### Experimental design for the creation of topological spin textures

Figure 1a shows the schematic diagram of the multilayers. The Hall bar is 20 μm in width and 120 μm in length (Sample fabrication details are given in the Method section). In our previous work, magnetic skyrmions have been generated with a current in the same stack[39,40]. Please refer to Supplementary Video 1 for more detailed information on the process of skyrmion creation. We find that this system can also be utilized to generate and stabilize various intriguing magnetic topological textures, including skyrmioniums. Fig. 1b shows the spin configurations of $Q = 1$ skyrmion, $Q = 0$ Néel-type skyrmionium, and $Q = -1$ skyrmion. Skyrmioniums are donut-like magnetic topological textures with an out-of-plane spin distribution[31]. The violet arrows connect the chiral domain wall with the same orientation between the skyrmion and the skyrmionium, indicating that the skyrmionium is formed by two skyrmions with opposite $Q$. It has been theoretically demonstrated that skyrmioniums have higher energy than skyrmions in general ferromagnetic systems and are susceptible to thermal fluctuations and perturbation caused by a current[31]. We therefore need to first create high-energy transitional states for creating skyrmioniums. Fig. 1c presents a series of magnetic optical Kerr effect (MOKE) images that depict the sequential evolution from initial skyrmions to subsequent formations of skyrmion bags, followed by the emergence of skyrmioniums, and ultimately transitioning into single $Q = -1$ skyrmions. Notably, we have demonstrated the ability to convert a skyrmion bag into individual skyrmions through the application of either square pulses or sinusoidal pulses. The transformation of the skyrmion bag into a skyrmionium exhibits distinct behaviors under the influence of square pulses and sinusoidal pulses. However, the key factor that drives these two different transformations is the variation in

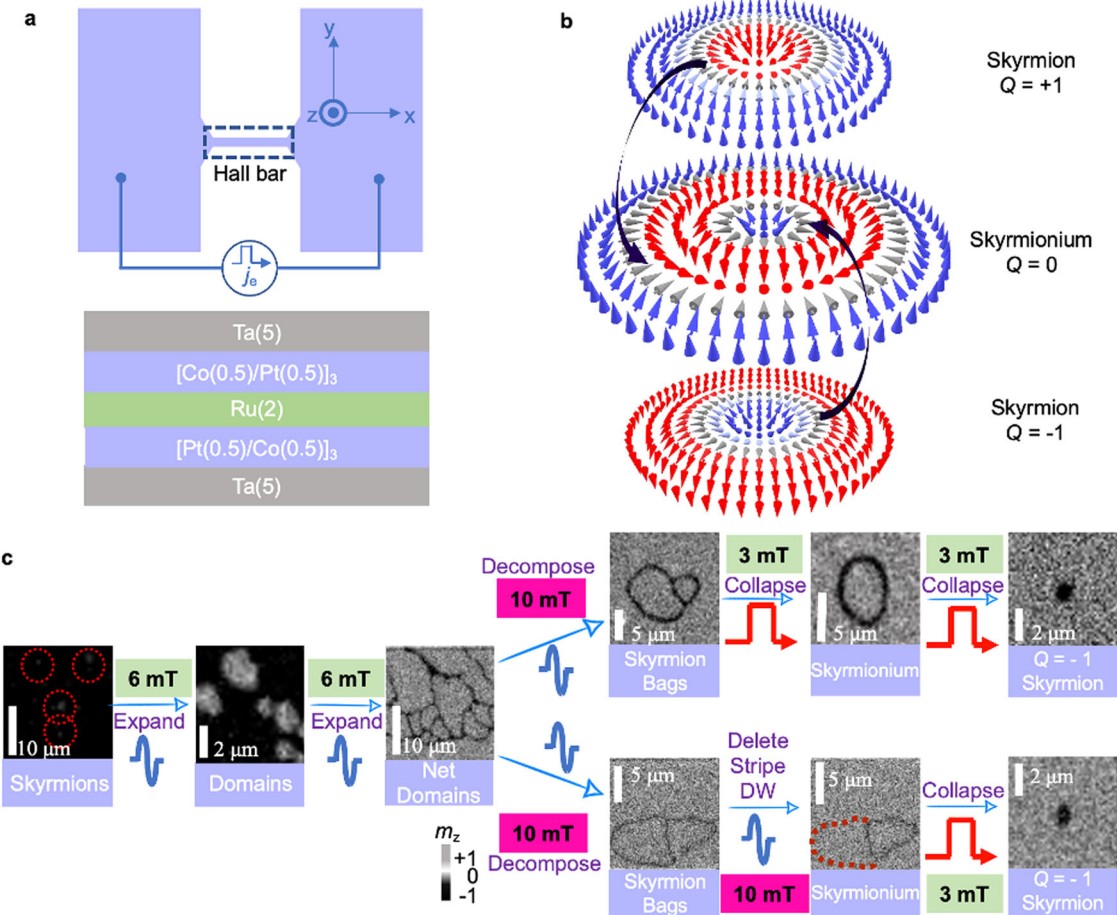

**Fig. 1 | Experimental design for the conversion between skyrmion and skyrmionium. a** Schematic diagram of the Ta(3)/[Pt(0.5)/Co(0.5)]₃/Ru(2)/[Co(0.5)/Pt(0.5)]₃/Ta(3) multilayers. Numbers in parentheses represent thicknesses in nanometers. The sequent current pulses are injected into the Hall bar along +x by default. The Hall bar is 20 μm in width and 120 μm in length. **b** The schematic

diagrams of a $Q = 1$ skyrmion, a $Q = 0$ skyrmionium and a $Q = -1$ skyrmion. The violet arrows connect the chiral domain wall with the same orientation between the skyrmion and the skyrmionium. **c** The evolution from skyrmions to skyrmion bags, and then to a $Q = -1$ skyrmion.

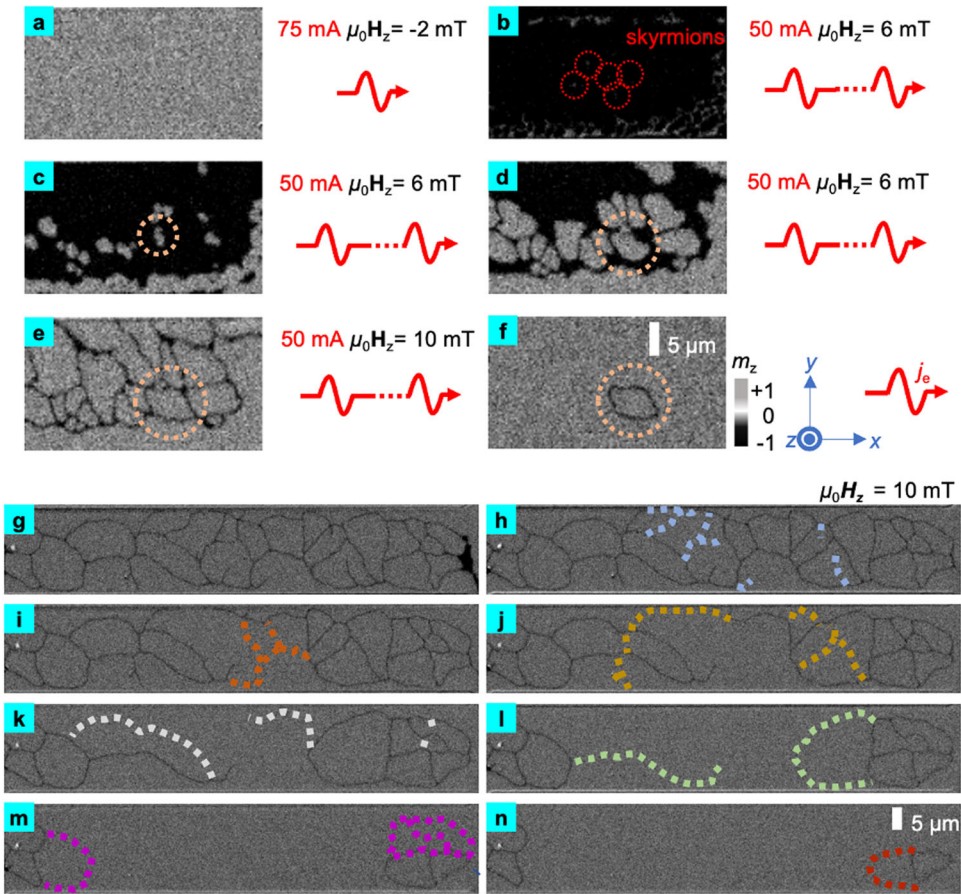

**Fig. 2 | The creation process of an isolated skyrmionium. a–f** The creation process of the isolated skyrmionium. The sinusoidal pulses are injected into the Hall bar along +*x* by default. **g–n** The transition from the stripe domains to the isolated skyrmionium. The dashed line represents the stripe domains that disappear after the sinusoidal pulse.

the external fields. We provide comprehensive discussions on the detailed experimental findings related to this phenomenon in the subsequent sections of this paper. Our experimental observations reveal that the skyrmioniums in our study exhibit a range of sizes, with diameters ranging from 1 to 15 µm. (Please refer to Supplementary Fig. 4 and Supplementary Video 9 for more details). In the MOKE image, the lighter gray scales correspond to the magnetization in the +z direction, while the darker gray scales represent the magnetization in the −z direction.

## Creation of skyrmionium

Through a series of sequential steps, as depicted in Fig. 2a–f, we have successfully achieved repeated generation of skyrmioniums in our sample. It would be helpful to provide a definition for the newly introduced magnetic structure, namely the "net domain". The net domain, initially presented in Fig. 2e and subsequently in Fig. 2g–l, is a collection of stripe domains that forms a complete fishnet-like pattern and is distributed throughout the device. After the decomposition of the net domain, the formation of skyrmion bags becomes evident. Their initial appearance is observed in Fig. 1c, followed by their presence in Fig. 2m, and a detailed discussion is provided in Fig. 3a–e. To generate skyrmioniums, we first saturate the sample with an external magnetic field of +200 mT for initialization, as shown in Fig. 2a. We then set the field to be −2 mT and injected a single sinusoidal pulse along +*x* with amplitude of $I_{sin} = 75$ mA (i.e., current density of $2.68 \times 10^{11}$ A/m²) into the Hall Bar. The magnetization of the Hall bar is switched from +z to −z after the stimulation of a single sinusoidal pulse, as shown in Fig. 2b, and magnetic skyrmions are created at the same time.

Please refer to Supplementary Video 2 for more details. Due to the short effective time of a sinusoidal pulse, the skyrmions exhibit minimal movement under its influence. As a result, pushing the skyrmions forward can only be achieved using a square pulse. For detailed information, please refer to Supplementary Fig. 3 and Supplementary Video 8. These skyrmions serve as the initial seeds for the subsequent formation of a net domain.

The third step is to generate the net domain by expanding the skyrmions. We set the external field to +6 mT, and the skyrmions inside the sample do not change significantly. We then continuously inject sinusoidal pulses with $I_{sin} = 50$ mA into the Hall bar, and the skyrmions gradually expand into domains until the shape of the magnetic topological textures (net domains) no longer fluctuate significantly, as shown in Fig. 2c, d. Under continuous stimulation of the sinusoidal pulses, the original domains with magnetization in −z are compressed into the net domain, as shown in Fig. 2e.

Finally, we decompose the net domain to create skyrmion bags and skyrmioniums. We increase the perpendicular field to 10 mT, and repeated the sinusoidal current pulses with $I_{sin} = 50$ mA. As shown in Fig. 2g–m, the width of the net domain with magnetization in −z is further compressed and start to decompose under the stimulation of the sinusoidal pulses. The net domain is soon torn apart into separated skyrmion bags, as shown in Fig. 2m, where the compressed net domain can be regarded as a combination of multiple stripe domains. The conjunction points of the different stripe domains form nodes. Please refer to Supplementary Video 3 for details.

To understand the decomposition process of the net domain, the obliterated stripe domains are marked with dashed lines. As depicted

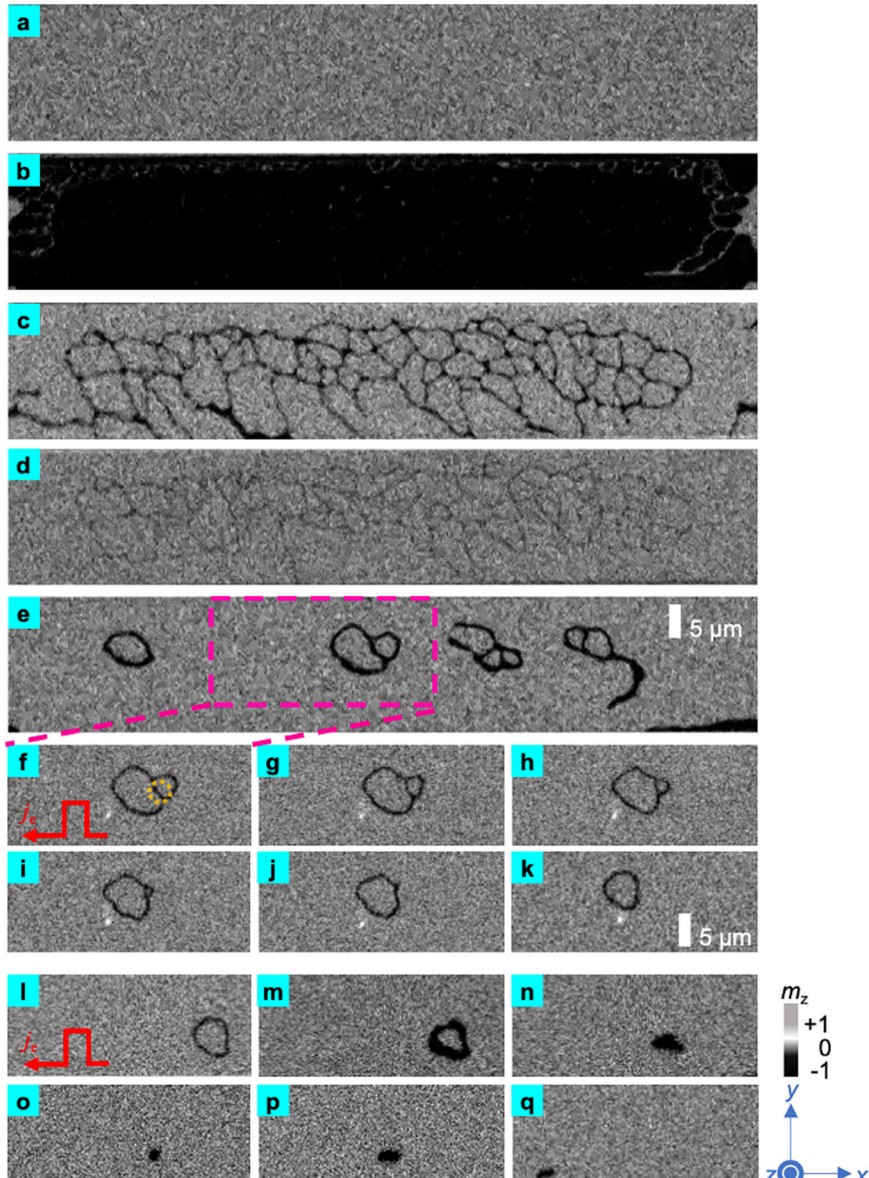

**Fig. 3 | Conversion of the topological textures. a–e** The creation process of the skyrmion bag. The creation process is identical to the one we performed in Fig. 2. **f–k** The conversion from a skyrmion bag to a skyrmionium. Square pulses are used to set the skyrmionium and skyrmion into motion. **l–q** The conversion from a skyrmionium to a skyrmion. In **n–q**, the field is decreased from 3 to 1 mT, to prevent the skyrmion from being destroyed.

in Fig. 2h–m, the disappearance of the net domain is first observed in the middle section and then in the left and right parts of the Hall bar. This is because the geometry of the Hall bar (Fig. 1a) concentrates the electrons, producing a higher current density in the middle part. Interestingly, the obliteration of net domains consistently takes place between nodes of stripe domains. This means that these nodes are unstable under the stimulation of current pulses. We believe that the magnetic moments of the nodes can be complex and vulnerable to the effect of spin orbit torques (SOTs), especially when the width of the stripe domain is compressed by an external magnetic field of 10 mT. The obliteration of the stripe domain is most likely due to the destruction of the nodes. Stripe domains can be eliminated by the square pulses, but it results in uncontrollable stripe domain destruction, often leading to the total decomposition of net domains. We use the sinusoidal pulses to stimulate the magnetic texture because it has a shorter effective time than the square pulses. (Please refer to Supplementary Fig. 3 for details). As we mentioned earlier in the paper, we can

observe a skyrmion bag on the right side of Fig. 2m. Continuous application of the sinusoidal pulses breaks the left part of the skyrmion bag and created an isolated skyrmionium, as shown in Fig. 2f, n. In general, we first excite the magnetic multilayer to a high energy translational state through the combined effect from the current and the external magnetic field. After the magnetization switching, the external field no longer competes with the magnetic anisotropy, and the moderate stimulation from sinusoidal pulses gradually relaxes the system to low energy ground state. During the relaxation, we sequentially observe the transformation of the net domain into skyrmion bag, skyrmionium and skyrmion, which can be regarded as footprints of the system in minimizing its total free energy. We perform micromagnetic simulations to demonstrate this process, which well reproduce our experimental observations, as shown in Supplementary Fig. 5. Further analyses on the energy state the of the observed magnetic structures also confirm the experiments, as shown in Supplementary Fig. 6.

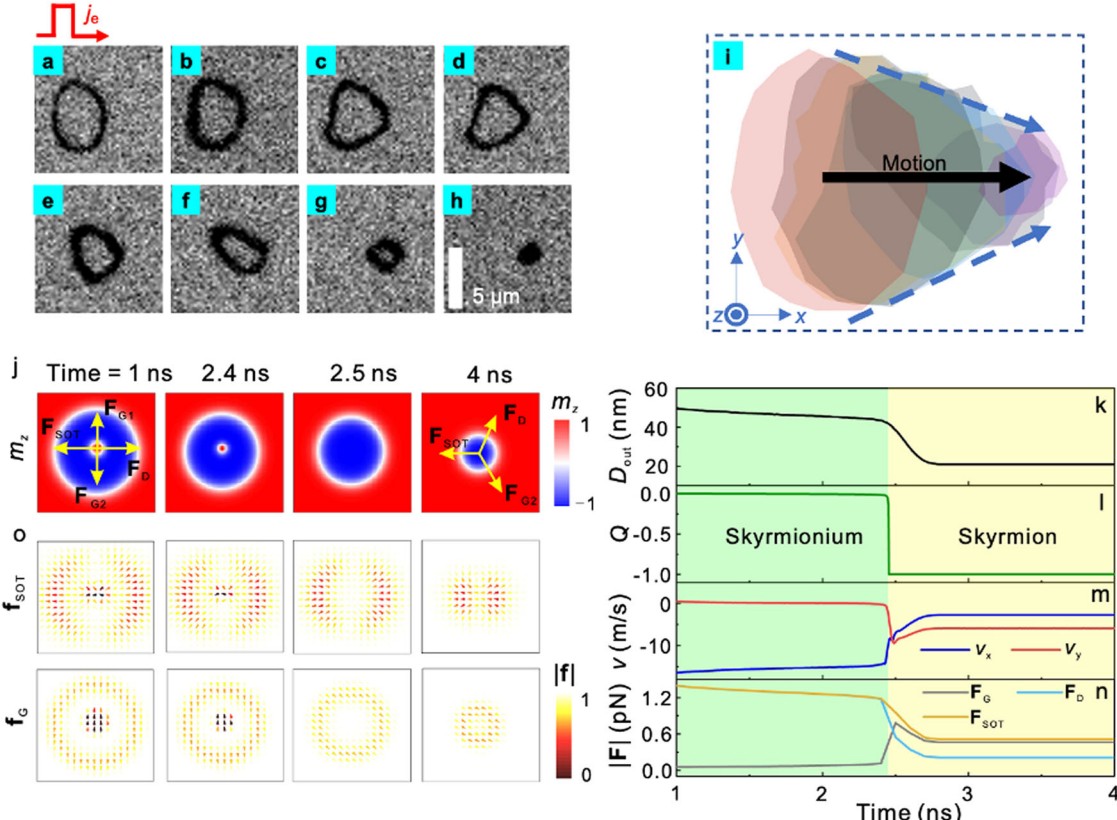

**Fig. 4 | The process whereby the skyrmionium collapses into a skyrmion with applied current. a–h** Current-driven motion of a skyrmionium. Sequent square pulses with an amplitude $I_{sq} = 50$ mA, and a pulse length of 50 μs are applied along $+x$. The out-of-plane field remains at 3 mT throughout the process. **i** The trajectory of the current-driven skyrmionium. The arrow points from the initial to the final position of the skyrmionium center, indicating that this center moves strictly along the current direction and does not exhibit the Hall effect. **j** Snapshots of the simulated magnetization distribution taken at 1, 2.4, 2.5 and 4 ns during the process in which the skyrmionium collapses into a skyrmion with applied current. **k–n** Time dependence of the outer diameter $D_{out}$, the topological number $Q$, the velocity and the effective forces of the spin structure. **o**, Distribution of the normalized effective force density of the spin orbit torque, $\mathbf{f}_{SOT}$ (upper panel) and the Magnus force, $\mathbf{f}_G$ (lower panel) corresponding to the spin configurations in (**j**). For the simulation, we assume the DMI constant $D = 3.38$ mJ/m$^2$, the current with a density of $J = 1.5 \times 10^{11}$ A/m$^2$ is applied in $-x$ direction, and the spin hall angle $\theta_{sh}$ of the Ta channel is 0.1.

## Conversion from skyrmion bag to skyrmionium

In addition to demonstrating the process described above, Fig. 3a–q also show another way of transforming the skyrmion bag into a skyrmionium. Theoretically, skyrmion bags with arbitrary integer topological number could be stabilized in a conventional micromagnetic model with Dzyaloshinskii Moriya interaction (DMI)[41]. These cluster-like structures have been recently observed in liquid crystal[38] and exfoliated flakes of FeGeTe[42]. According to the polarity of the isolated skyrmions we observed in experiment, the skyrmion bags has a topological number $Q = +1$. Fig. 3a–e demonstrate the creation process of the skyrmion bag, which is identical to the process of Fig. 2a–m. To observe its evolution, we set the perpendicular field to 3 mT and applied square pulses along $-x$ with $I_{sq} = 50$ mA. The field here is set to be 3 mT, which differs from the earlier decomposition process set to be 10 mT. Please refer to the Supplementary Fig. 3 for details. Fig. 3e shows the initial state of a skyrmion bag. The creation of a skyrmion bag follows a similar process to that of a skyrmionium. When pulses are injected into the device, the right portion of the skyrmion bag starts to shrink, as shown in Fig. 3g–i. In Fig. 3j. The right part of the skyrmion bag collapses to a dot, and it has disappeared completely as shown in Fig. 3k. As shown in the sequence Fig. 3f–k, the topological number $Q$ drops from 1 to 0 when the skyrmion bag converts into a single skyrmionium. We further perform micromagnetic simulation to demonstrate the possibility to stabilize skyrmion bags in the magnetic system under investigation, as presented in Supplementary Fig. 5.

## Conversion from skyrmionium to skyrmion

The process of converting a skyrmionium to a skyrmion is shown in Fig. 3l–q. Fig. 3l shows the initial state of a skyrmionium. As the square pulse is injected into the device, the skyrmionium shrinks and its inner skyrmion eventually collapses, as shown in Fig. 3m, n. This phenomenon has been reported in previous theoretical work[31]. After the skyrmionium collapses into a skyrmion, the skyrmion can be pushed towards $-x$ by square pulses (The field is decreased to 1 mT to prevent the skyrmion from being destroyed). However, since the topological number of the skyrmion is $-1$, it will experience the SkHE and move towards the edge of the device, as shown in Fig. 3n–q. At this stage of the process, we observe a topological transformation of the spin textures. Please refer to Supplementary Videos 4–6 for more details.

## SkHE free current-driven skyrmionium motion

In the previous section, we discuss the conversion between magnetic topological textures with different topological numbers. Among these textures, the skyrmionium stands out due to its topological number being zero. Consequently, when the skyrmionium is driven by an electric current, we would expect no significant SkHE to occur. Figure 4a–h indeed shows the SkHE-free motion of the skyrmionium. As shown in Fig. 4a, when only a perpendicular field of 3 mT is applied, the skyrmionium will not display any observable deformation or displacement, indicating that it is stable under this external field condition. We then continuously inject the square pulse with the amplitude

$I_{sq} = 50$ mA into the device along $+x$. We have observed that the skyrmionium undergoes a gradual collapse, transforming into a skyrmion and eventually being obliterated, as illustrated in Fig. 4b–h. Please refer to Supplementary Video 7 for additional details. The behavior of the skyrmionium strongly depends on the field and current amplitude. For more information, please refer to Supplementary Fig. 2. The pulse widths are in the range of microseconds, so we hypothesize that the heat dissipation may act as a catalyst, assisting the spin-orbit torque (SOT) to overcome the pinning barrier and set the skyrmionium into motion.

As we take a closer look at the current-driven dynamics of the spin textures, we discover both the similarities and the differences between skyrmioniums and skyrmions. As shown in Fig. 4b, after the square pulse is applied, the ring of a skyrmionium becomes thicker. A similar change of size has also been observed in skyrmions[39,40]. From Fig. 4b–g, it can be observed that the diameter of the skyrmionium gradually decreases, whereas the thickness of the ring does not change significantly. Under the continuous stimulation of the current pulses, the skyrmionium eventually collapses into a skyrmion, as shown in Fig. 4h. However, since this time the field does not decrease to 1 mT, as in our previous observations (Fig. 3n–q), the applied field destabilizes the skyrmion, and the skyrmions will soon be obliterated. Fig. 4i gives a schematic snapshot of Fig. 4a–h. This snapshot provides a record of the relative position and size of the skyrmionium. Notably, we observe that the skyrmionium undergoes both a collapse in size and a displacement caused by the current. Interestingly, the motion of the skyrmionium is not affected by the Skyrmion Hall Effect (SkHE), as indicated by the black arrow in Fig. 4i, pointing from the initial center point to the final collapse point of the skyrmionium. This observation aligns with theoretical expectations, as the skyrmionium has a topological number of 0 and does not exhibit a velocity perpendicular to the direction of the current as it is driven by the current[43,44].

## Micromagnetic simulations

To validate our experimental findings, we conduct additional micromagnetic simulations. Specifically, we assume a DMI constant below the threshold value required for the stabilization of the skyrmionium. We observe the process of a pre-set skyrmionium being excited by SOT, gradually collapsing and transforming into a skyrmion. Figure 4j shows the snapshots of the magnetization distribution at critical time points during this process. The diameter of the skyrmionium decreases while it moves along the direction of the current. At approximately 2.5 ns, the inner core of the skyrmionium is obliterated, resulting in a significant change in its size. Subsequently, a stable skyrmion with a consistent diameter emerges. This progression can be observed in Fig. 4k. The topological property and the dynamics of the spin textures also change during the process. Before the collapse, the skyrmionium moves along the $-x$ direction at a speed of 15 m/s, and the skyrmion Hall effect is absent (Fig. 4m). The annihilation of the inner core of the skyrmionium results in a sudden change of the topological number from 0 to $-1$ (Fig. 4l). The change in topology upsets the balance of the Magnus forces experienced by the spin texture, and produces the translational drift, indicated by a non-zero speed in $-y$ direction as denoted by the red solid line in Fig. 4m. This process qualitatively captures the physics underlying our experimental results, and provides convincing evidence for the chirality of the observed spin structures.

## Discussion

By projecting the Landau-Lifshitz–Gilbert (LLG) equation onto the translational mode that incorporates SOT, the dynamics of the spin structure can also be described by Thiele's equation:

$$\mathbf{F}_G + \mathbf{F}_D + \mathbf{F}_{SOT} = \mathbf{0} \tag{1}$$

where $\mathbf{F}_G = \mathbf{G} \times \mathbf{v}$ is the Magnus force, with $\mathbf{G} = -4\pi Q \mu_0 M_s t \gamma^{-1} \mathbf{e}_z$ being the gyro-vector, and $\mathbf{v}$ being the drift velocity of the spin structure. $\mathbf{F}_D = -\alpha \mu_0 M_s t \mathbf{d} \cdot \mathbf{v}/\gamma$ is the effective force of magnetic damping, and $\mathbf{d}$ is the dissipative tensor with the component $d_{ij} = \int dx dy \left( \partial_i \mathbf{m} \cdot \partial_j \mathbf{m} \right)$. $\mathbf{F}_{SOT} = -\mu_0 H_J M_s t \mathbf{u}$ is the effective force of spin orbit torque, with $H_J = J\hbar\theta_{sh}/(2\mu_0 e M_s t)$ and $u_i = \int dx dy \left[ (\mathbf{m} \times \mathbf{p}) \cdot \partial_i \mathbf{m} \right]$ when only the damping-like component is considered. In the above definitions, $t$ is the thickness of the magnetic layer, $M_s$ is the saturation magnetization, $\mu_0$ is the vacuum permeability, $\gamma$ is the gyromagnetic ratio, $\alpha$ is the damping constant, $J$ is the current density, $e$ is the elementary charge, $\theta_{sh}$ is the spin Hall angle, $\hbar$ is the reduced Planck constant, $\mathbf{m}$ is the reduced magnetization vector, and $\mathbf{p}$ is the polarization direction of the current. Utilizing the spin configurations and the velocity obtained by micromagnetic simulations, the force experienced by the spin structure can be analyzed. For a ferromagnetic skyrmionium, the major contribution to the driving force comes from both the inner and the outer domain wall being perpendicular to the current, as shown in the upper panel of Fig. 4o. The annihilation of the inner core of the skyrmionium wipes out part of the spin textures that capture the angular momentum from the polarized current, and decreases the magnitude of $\mathbf{F}_{SOT}$ from 1.2 pN to about 0.5 pN, as shown by the orange line in Fig. 4n. However, the degeneration of mobility during the collapse of the skyrmionium has not been observed in our experiments. This might be due to the pinning effects in our sample. The lower panel of Fig. 4o shows the distribution of the Magnus force corresponding to the spin configurations in Fig. 4j. As predicted by our previous studies[31], the inner and outer domain wall of the skyrmionium possess opposite topological numbers, leading to a zero net Magnus force $\mathbf{F}_G$, as indicated by the grey line in Fig. 4n. Upon the collapse of the skyrmionium, the Magnus force of the inner core $\mathbf{F}_{G1}$ vanishes, and that of the outer domain $\mathbf{F}_{G2}$ is rebalanced by $\mathbf{F}_{SOT}$ and $\mathbf{F}_D$. The forces experienced by skyrmioniums and skyrmions are indicated by the yellow arrows in Fig. 4j. This discussion demonstrates that higher mobility and the absence of the skyrmion Hall effect are the two major advantages possessed by skyrmioniums over skyrmions. We also want to highlight that the dynamic process of the skyrmionium before annihilation is quasistatic, and the Thiele's equation is applicable in this context.

In summary, we have presented two approaches for converting a skyrmion into a skyrmionium, utilizing the skyrmion bag as the transitional state. The first approach involves generating a net domain from the skyrmion using square pulses and subsequently decomposing it to form skyrmioniums by reducing the topological charge to zero. The second approach focuses on collapsing skyrmion bags into skyrmioniums using square pulses. Our findings demonstrate the feasibility of interconverting between skyrmion bags, skyrmions, and skyrmioniums. Furthermore, we utilize this conversion mechanism to successfully create a skyrmionium and investigate its current-driven motion, which is free from the effects of the Skyrmion Hall Effect (SkHE). These findings have significant implications for overcoming critical challenges in the development of skyrmionics devices for practical applications.

## Methods

### Experimental method

The multilayer stack Ta(3)/[Pt(0.5)/Co(0.5)]₃/Ru(2)/[Co(0.5)/Pt(0.5)]₃/Ta(3) (thicknesses are in nanometers; refer to the inset in Fig. 1a for a cross-sectional view) is deposited onto a Si substrate with a 300-nm-thick oxide layer at room temperature using a high-vacuum magnetron sputtering system (AJA International Inc.). The base pressure of the vacuum system is below $4 \times 10^{-6}$ Pa. The thickness of Ru $t_{ru} = 2$ nm is precisely tuned to induce the ferromagnetic coupling of the upper and lower Pt/Co magnetic layers. Using lift-off photolithography, the multilayer is patterned to a 20-µm-wide device, which is attached to a homemade printed circuit board and bonded with electrodes for the

injection of different pulses (Keithley 6221 current source). All the current densities are of the order of $10^{11}$ A/m$^2$. Our previous studies have indicated that the Co layers have a PMA[37]. The bottom Ta layer serves as a seed layer for the nucleation of top adlayers and also as a spin-current injector to generate SOT for skyrmionium manipulation. A MOKE microscope is used for direct imaging.

## Computational method

We use the open-source micromagnetic code pack Mumax3[45] for the modeling of the ferromagnetic system. The parameters used for the simulations are adopted from Pt/Co/Ru/Co/Pt multilayers, with the saturation magnetization $M_s = 1 \times 10^6$ A/m, exchange constant $A_{ex} = 1.5 \times 10^{-11}$ J/m, and perpendicular magnetic anisotropy constant $K_u = 1.3 \times 10^6$ J/m$^3$. The simulated region is 512 nm × 512 nm × 1 nm with a mesh size of 1 nm × 1 nm × 1 nm. For the above parameters, the skyrmionium is not stable with DMI constant below a threshold of 3.5 mJ/m$^2$, as shown in Supplementary Fig. 1. In this case, the skyrmionium observed in our experiments could be in a transient state fixed by the pinning effects, and it tends to collapse once excited by spin-orbit torque or thermal fluctuations caused by the current flows in the bottom Ta layer. We assume the DMI constant $D = 3.38$ mJ/m$^2$ for the simulations. In addition, the thickness of Co layer in our sample is 0.5 nm, for which we assume a large damping constant $\alpha = 0.3$. On the other hand, the thickness of the Co layer is much smaller than its electron mean free path, resulting in very large resistivities in the magnetic layers. Thus, the effect from spin-transfer torque in our system can be ignored.

## Data availability

The authors declare that the data supporting the findings of this study is available within the article and its Supplementary Information file. All other data that support the results of this study are available from the corresponding author upon reasonable request.

## Code availability

The source code of Mumax3 is available at https://mumax.github.io/.

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

## Acknowledgements

This work was supported by funding from the Guangdong Special Support Project (2019BT02X030), the Shenzhen Peacock Group Plan (KQTD20180413181702403), the Shenzhen Fundamental Research Fund (Grant No. JCYJ20210324120213037), the Guangdong Basic and Applied Basic Research Foundation (2021B1515120047), the National Natural Science Foundation of China (11974298) and Shenzhen Key Laboratory of Functional Aggregate Materials (ZDSYS20211021111400001). Y.L.Z. acknowledges funding support from the National Natural Science Foundation of China (Grant No. 12004319). Z.Q.C. acknowledges financial support from the HKSAR Research Grants Council (RGC) Early Career Scheme (ECS; No. 27202919); the HKSAR Research Grants Council (RGC) Research Matching Grant Scheme (RMGS, No. 207300313); and the Guangdong Special Support Project (No. 2019BT02X030). X.X. acknowledges financial support from the National Natural Science Foundation of China (51871137 and 61434002) and the National Key R&D Program of China (2017YFB0405604). X.L. acknowledges financial support from the National Natural Science Foundation of China (Grant No. 12104322) and the Shenzhen Science and Technology Program (Grant No. ZDSYS20200811143600001). J.Å. acknowledges financial support from the Swedish Research Council (VR; 2017-06711 and 2019- 04229).

## Author contributions

Y.Z. conceived the work. Y.Z., J.A., and X.G.L supervised the project. X.G.L. carried out the simulation. S.Y., Y.L.Z., and K.W. performed the experiments under the supervision of Y.Z. All authors participated in discussions regarding the results and contributed to the preparation of the manuscript.

## Competing interests

The authors declare no competing interests.
