## [Peer Review File · Nature Communications]

Reviewers' Comments:

Reviewer #1:

Remarks to the Author:

Yang et al experimentally present multiple magnetic textures (skyrmion, skyrmionium, net domain, multi-Q domain) and their transformation via an electric current and magnetic field in a multilayered thin film system using MOKE observation. The authors claim two ways to drive such conversion by sinusoidal and square pulses, respectively, in a phenomenological way. However, the writing description of the experimental process is confusing and difficult to understand. Secondly, I cannot see the intrinsically different mechanisms for the two approaches; what is the role of the sinusoidal pulse and square pulse? Besides, the size of the domain is more than 1 μm , which is too big for the claimed spintronic application. So, It is difficult to recommend the publication in Nature Communications; an alternate journal may consider a substantially revised version.

Below are several issues:

1. The authors use the zero Hall effect to confirm that the observed texture is skyrmionium, which I can't entirely agree with because the size of the magnetic circle domain is too large, and the trace will be affected by other facts, such as material defects. Besides, why did the center of the observed contrast not show a dot, as there should exist up/downwards spins?
2. Can you define the winding number of the observed "multi-Q cluster"?
3. The author claims the magnetic domains in Fig.2c-d are skyrmions, which I doubt; they maybe just domain clusters.
4. It is difficult to understand "the topological charge will not be the same as for two individual skyrmioniums." for Fig. 3
5. For Fig. 1: the left one of (c) cannot be defined as "skyrmions"; they are just magnetic domain clusters. Please add the scale bar on each image. Since (c) process is experimental data, please add two images for the final states, although they have similar features maybe.

Reviewer #2:

Remarks to the Author:

Key results

The authors describe the experimental creation of skyrmions and skyrmioniums on a film of Ta(3)/[Pt(0.5)/Co(0.5)]₃/Ru(2)/[Co(0.5)/Pt(0.5)]₃/Ta(3) multilayers. Skyrmions are created, they form clusters that can be turned into skyrmioniums, and turned again into skyrmions, through the action of voltage pulses.

Validity

The results appear to be robust.

Significance

The possibility of producing skyrmions and skyrmioniums in a reproducible way at room temperature is very important, both for the investigation of the properties of these systems, and for the development of spintronic applications.

Data and methodology

The presentation is in general clear, and the experimental techniques used are appropriate to obtain the desired nanoscopic structures.

Analytical approach

The analysis of the experiments described in the paper appears to be robust.

Suggested improvements:

a) Comments and suggestions for the Abstract.

Line 40: The authors state: "The skyrmion Hall effect of skyrmioniums has been suppressed to zero, which supports that the winding number of skyrmioniums indeed is reduced to zero."

It is not necessary for the paper to suggest that one of their findings is that the skyrmionium has zero winding or skyrmion number; this is a known fact. It would be more adequate to state:

Since the Skyrmion Hall effect is reduced to zero, this confirms that the particles are skyrmioniums, known to have $Q=0$.

Line 44: Still in the abstract, the authors state:

"The establishment of reversible conversions between different magnetic topological spin textures is an important development, which should speed the advent of the next generation of spintronic device."

Since the bibliography contains articles that have already studied theoretically the feasibility of such conversions, my suggestion is to change to

The experimental establishment of...spintronic devices.

b) Comments and suggestions for the main text

Line 56: Skyrmioniums, with topological winding number $Q=0$, ...

It is better to state clearly that you cannot have skyrmioniums with Q different from 0:
Since Skyrmioniums have topological winding number $Q=0$,

Line 59: However, the Néel-type skyrmion or skyrmionium is considered a better information carrier...

If here the authors are identifying the Néel-type skyrmion with the skyrmionium they are wrong!
To avoid the risk of misleading the reader, I suggest:

The skyrmionium is considered a better information carrier...

Line 66: However, the conversion between skyrmions and skyrmioniums is more complicated and challenging, involving multiple conversions of different magnetic structures...

The experimental work described here does indeed describe conversions involving several steps, but since some theoretical works in the bibliography describe single-step conversions, maybe the authors should point out this difference.

Line 321: In j-l, the field is decreased from 3 mT to 1 mT prevent skyrmion from being destroyed.

In j-l, the field is decreased from 3 mT to 1 mT, to prevent the skyrmion from being destroyed.

Clarity and context

The procedures and results obtained are well described.

References

I think the relevant literature is reviewed properly.

Providing constructive feedback

I have made my suggestions for the authors in the "Suggested improvements" section.

Response to Reviewer 1's Comments

Reviewer #1:

Yang et al experimentally present multiple magnetic textures (skyrmion, skyrmionium, net domain, multi-Q domain) and their transformation via an electric current and magnetic field in a multilayered thin film system using MOKE observation. The authors claim two ways to drive such conversion by sinusoidal and square pulses, respectively, in a phenomenological way. However, the writing description of the experimental process is confusing and difficult to understand. Secondly, I cannot see the intrinsically different mechanisms for the two approaches; what is the role of the sinusoidal pulse and square pulse? Besides, the size of the domain is more than 1 μm , which is too big for the claimed spintronic application. So, it is difficult to recommend the publication in Nature Communications; an alternate journal may consider a substantially revised version.

1.The authors claim two ways to drive such conversion by sinusoidal and square pulses, respectively, in a phenomenological way. However, the writing description of the experimental process is confusing and difficult to understand.

Authors' Reply:

We would like to express our gratitude to the reviewer for their helpful feedback. To enhance the readability of our manuscript and prevent any ambiguity, we have meticulously revised the figures and rephrased the discussion related to the experimental process. Our modifications have resulted in a more coherent and comprehensible depiction of the figures, making it easier for readers to understand the content.

Fig. 1c presents a schematic representation of the conversion of multi-Q

clusters (now referred to as skyrmion bags in the revised manuscript) into skyrmioniums through two distinct methods - sinusoidal and square pulses. However, it only depicts the conversion process without indicating the external field information, which may lead to confusion that the waveform alone is responsible for the conversion. Therefore, we acknowledge that additional details are required to fully comprehend the conversion process.

We have taken the reviewer's feedback into consideration and made significant improvements to Fig. 1c and the main text to enhance their clarity. Our revisions emphasize that the combination of external fields and waveforms, rather than the waveform alone, is crucial for triggering the two distinct methods of creating skyrmionium, as illustrated in Fig. 1c. We have mentioned "*The field here is set to be 3 mT, which differs from the earlier decomposition process set to be 10 mT, please see the supporting material S3 for details.*" in line 173 of the revised manuscript.

We have included additional results in our reply to explain the reason for using different waveform pulses in each conversion process. We believe that these details provide a better understanding of our methodology and results. Please refer to our response below for more information.

2.what is the role of the sinusoidal pulse and square pulse?

Authors' Reply:

In general, sinusoidal pulses are utilized to expand skyrmions and decompose the net domain, while square pulses are used to initiate the motion of magnetic textures. As shown in Fig. 1c, there are two methods for generating a skyrmionium: by using either square pulses or sinusoidal pulses. However, it is important to note that the external field's influence is significant in the motion or conversion of a skyrmion bag. The creation process involves multiple steps,

which are outlined below:

Step 1: To generate skyrmions within the device, sinusoidal pulses (75 mA, 50 μ s) were employed while setting the external field to -2 mT.

Step 2: To expand skyrmions into the net domain, sinusoidal pulses (50 mA, 50 μ s) were utilized while setting the external field to 6 mT. Supporting material S3 and a supplemental video (Video S3) were added to elucidate the rationale behind the use of sinusoidal pulses in step 2. In the first part of the video, the skyrmions gradually expand under a 6 mT external field, with either full or half sinusoidal pulses applied. In contrast, square pulses instantly switch the magnetization, leaving behind only a few stripe domains. Full and half sinusoidal pulses function similarly, suggesting that sinusoidal pulses offer better control due to their shorter effective time, as demonstrated in supporting material S3.

Step 3: To decompose the net domain into a skyrmion bag, sinusoidal pulses (50 mA, 50 μ s) were applied while setting the external field to 10 mT. The second part of Video S3 highlights a critical difference between sinusoidal and square pulses. While sinusoidal pulses decompose the net domain gradually, square pulses perform the same operation in a single step. This phenomenon is attributed to the longer effective time of the square pulse, as depicted in Fig. R1. Once the skyrmion bag has been created, two methods are available to generate a skyrmionium. First, the skyrmion bag can be pushed by a square pulse (50 mA, 50 μ s, external field set to 3 mT), resulting in the shrinkage of the skyrmion bag to a skyrmionium. Second, sinusoidal pulses (50 mA, 50 μ s, external field set to 10 mT) can be continuously used to decompose the skyrmion bag into a skyrmionium. Finally, the skyrmionium can be converted into a skyrmion by applying square pulses (50 mA, 50 μ s, external field set to 1 mT).

The stability of a magnetic texture varies with different external fields, so we set the external field to 10 mT. As depicted in Fig. R1, we use sinusoidal pulses to stimulate the magnetic texture because they have a shorter effective time than square pulses, making them easier to control. The formation of a skyrmionium involves several steps, each requiring a specific external field amplitude and waveform. The primary objective of this study is to investigate how to achieve the conversion of a skyrmion to a skyrmionium and subsequently to a skyrmion with the opposite topological charge.

Fig. R1. The configurations of different waveforms.

3. Besides, the size of the domain is more than 1 μm , which is too big for the claimed spintronic application

Authors' Reply:

The reviewer's comment on the size of the magnetic textures studied in the present work is valid, as they may be too large for state-of-the-art spintronic device applications. However, it should be noted that smaller skyrmions may be difficult to image using conventional techniques such as MOKE microscopy.

The size of skyrmions is influenced by parameters such as magnetic anisotropy and DMI in theoretical models, while in experimental studies, the size can be adjusted by factors such as the magnetic stack structure and external magnetic field. As demonstrated in Fig. R2, smaller skyrmioniums ($\sim 1.3 \mu\text{m}$) can be created using the same method mentioned in our manuscript. However, we did not include these results in the main text as their inner domain is difficult to identify with current imaging methods. We chose larger skyrmioniums to study their current-driven dynamics, but we plan to optimize the magnetic stack for the electrical detection of smaller skyrmioniums using techniques like TMR. Yao Guang et. al [*Advanced Electronic Materials* 9 (1), 2200570 (2023)] has already demonstrated the electrical detection of skyrmions in magnetic tunnel junctions. In our manuscript, we primarily focus on the conversion between skyrmions and skyrmioniums and their dynamics.

Fig. R2 **a-b** Skyrmioniums creation in Ta(3)/[Pt(0.5)/Co(0.5)]₃/Ru(2)/[Co(0.5)/Pt(0.5)]₃/Ta(3). **c** Skyrmionium creation in Ta(3)/[Pt(0.5)/Co(0.6)]₃/Ru(2)/[Co(0.6)/Pt(0.5)]₃/Ta(3). The skyrmionium has a relatively small diameter of about 1.30 μ m.

Below are several issues:

1. The authors use the zero Hall effect to confirm that the observed texture is skyrmionium, which I can't entirely agree with because the size of the magnetic circle domain is too large, and the trace will be affected by other facts, such as material defects. Besides, why did the center of the observed contrast not show a dot, as there should exist up/downwards spins?

Authors' Reply:

There seems to be a misunderstanding regarding the identification of skyrmionium. Firstly, we identify the skyrmionium based on its morphology and subsequently confirm its SkHE-free characteristics by performing current-driven dynamics. In regards to concerns about the potential influence of material defects on the motion of the skyrmionium, we posit that it is reasonable to assume that the distribution of defects within the sample is random. Nevertheless, our experimental findings indicate that the nucleation of skyrmionium can occur in various regions of the sample through the application of an electric current. These skyrmioniums consistently display SkHE-free motion when stimulated by an electric current and move in a direction opposing the applied current.

Our previous study [*Physical Review Applied*, 18(2), 024030, 2022] revealed that skyrmions can indeed be generated within the sample, and these structures exhibit SkHE motion when stimulated by an electric current. These observations led us to conclude that the defects present within the sample do not significantly contribute to the SkHE-free motion of the skyrmionium. The real-space MOKE image of the skyrmionium depicted in Fig. 1c displays a red ring texture and includes inner and outer domain walls, as represented by the skyrmionium illustration shown in Fig. 1b. We concur with the reviewer's comment that the inner domain of the skyrmionium is relatively expansive and

should be classified as a skyrmionium bubble.

2. Can you define the winding number of the observed “multi-Q cluster”?

Authors’ Reply:

In our experiments, we observed cluster-like spin textures that belong to the general definition of skyrmion bags [*Phys Rev. B* 99, 064437 (2019)]. This phenomenon has also been found in liquid crystals [*Nat. Phys.* 15, 655-659 (2019)] and van der Waals magnets very recently [*Adv. Mater.*, 2208930 (2023)]. Based on the polarity of the observed spin texture, the spin textures shown in Fig. 3e can be classified as skyrmion bags with a topological charge $Q = +1$. To demonstrate the possibility of stabilizing skyrmion bags in the magnetic system under investigation, we performed complementary micromagnetic simulations. Fig. R3 shows the simulated magnetization evolution of the magnetic thin film, which was relaxed from a random initial state. At a simulation time of about 5 ns, we observed the coexistence of skyrmion ($Q = -1$), skyrmionium ($Q = 0$), and skyrmion bag ($Q = 1$). We further analyzed the energy state of these magnetic structures to demonstrate their relative stability, as shown in Fig. R4. Generally, the topological charges of skyrmion bags have the highest energy cost, followed by the skyrmionium, while the skyrmion has the lowest energy cost. In our experiments, we observed the sequential transformation of the net domain into skyrmion bag, skyrmionium, and skyrmion. This process can be well explained by the relaxation of an excited magnetic system in minimizing its total free energy. We have added these results and corresponding descriptions to the supplementary material.

Fig. R3 Simulated magnetization evolution of the magnetic thin film relaxed from a random initial state. The snap shots are taken at the simulation time of **a** 0.05ns, **b** 0.75ns, **c** 1.75ns and **d** 5ns.

Fig. R4 Energy cost per topological charge of skyrmion, skyrmionium and skyrmion bag. **a** Total energy as a function of the perpendicular external field H_{\perp} . **b** Energy composition of the observed magnetic structures, including the

energy contributions from the magnetic anisotropy E_{anis} , Heisenberg exchange E_{exch} , Dzyaloshinskii–Moriya interaction E_{dmi} , and demagnetizing field E_{demag} .

3. The author claims the magnetic domains in Fig.2c-d are skyrmions, which I doubt; they maybe just domain clusters.

Authors' Reply:

Thank you for your comment. We would like to clarify that the domains shown in Fig. 2c-d were generated from the expansion of the original skyrmions and are likely no longer considered as skyrmions. Therefore, we have revised our description in the revised manuscript to avoid any confusion.

4. It is difficult to understand “the topological charge will not be the same as for two individual skyrmioniums.” For Fig. 3

Authors' Reply:

As shown in Fig. 3a, the two skyrmioniums are conjunct and share the same stripe domain, outlined with a circle in the figure. On the other hand, two separated skyrmioniums do not share the same stripe domain, resulting in a total topological number of zero. This main difference leads to different topological charges. Therefore, we have updated the terminology from "individual skyrmioniums" to "separated skyrmioniums".

5. For Fig. 1: the left one of (c) cannot be defined as “skyrmions”; they are just magnetic domain clusters. Please add the scale bar on each image. Since (c) process is experimental data, please add two images for the final states, although they have similar features maybe.

Authors' Reply:

In Fig. 1c, we replaced the original illustration on the left with a MOKE image of skyrmions to demonstrate the existence of two distinct methods for constructing a skyrmionium from individual skyrmions. The image incorporates scale bars for reference, and the final states of the conversion have also been added. It should be noted that the external field is the key to inducing the transformation, which is also shown in Fig. 1c. Additionally, the experimental details are primarily displayed in Figs. 2 and 3, as well as in the supplementary videos.

Response to Reviewer 2's Comments

Reviewer #2

Key results

The authors describe the experimental creation of skyrmions and skyrmioniums on a film of Ta(3)/[Pt(0.5)/Co(0.5)]₃/Ru(2)/[Co(0.5)/Pt(0.5)]₃/Ta(3) multilayers. Skyrmions are created, they form clusters that can be turned into skyrmioniums, and turned again into skyrmions, through the action of voltage pulses.

Validity

The results appear to be robust.

Significance

The possibility of producing skyrmions and skyrmioniums in a reproducible way at room temperature is very important, both for the investigation of the properties of these systems, and for the development of spintronic applications.

Data and methodology

The presentation is in general clear, and the experimental techniques used are appropriate to obtain the desired nanoscopic structures.

Analytical approach

The analysis of the experiments described in the paper appears to be robust.

Suggested improvements:

a) Comments and suggestions for the Abstract.

1. Line 40: The authors state: "The skyrmion Hall effect of skyrmioniums has been suppressed to zero, which supports that the winding number of skyrmioniums indeed is reduced to zero." It is not necessary for the paper to suggest that one of their findings is that the skyrmionium has zero winding or

skyrmion number; this is a known fact. It would be more adequate to state: Since the Skyrmion Hall effect is reduced to zero, this confirms that the particles are skyrmioniums, known to have $Q=0$.

Authors' Reply:

Thanks for the comment. We have deleted this sentence from the abstract, and stated that "*We have also directly observed the topological conversion from a skyrmionium to a skyrmion, characterized by the abrupt emergence of the skyrmion Hall effect.*" In line 40 (revised version).

2. Line 44: Still in the abstract, the authors state: "The establishment of reversible conversions between different magnetic topological spin textures is an important development, which should speed the advent of the next generation of spintronic device." Since the bibliography contains articles that have already studied theoretically the feasibility of such conversions, my suggestion is to change to The experimental establishment of...spintronic devices.

Authors' Reply:

We have changed "*The establishment of reversible conversions between different magnetic topological spin textures is an important development, we believe that our findings will assist the development of the next generation of spintronic devices.*" to "*The experimental establishment of reversible conversion between different magnetic topological spin textures is an important result that may accelerate the advent of next-generation spintronic device.*" in line 41 (revised version).

b) Comments and suggestions for the main text

1. Line 56: Skyrmioniums, with topological winding number $Q=0$, ...It is better

to state clearly that you cannot have skyrmioniums with Q different from 0:
Since Skyrmioniums have topological winding number $Q=0$,

Authors' Reply:

We have changed "*Skyrmioniums, with topological winding number $Q = 0$,*"
to "*Skyrmioniums always have a topological winding number $Q = 0$,*" in
line 54 of the revised manuscript.

2. Line 59: However, the Néel-type skyrmion or skyrmionium is considered a
better information carrier... If here the authors are identifying the Néel-type
skyrmion with the skyrmionium they are wrong! To avoid the risk of misleading
the reader, I suggest: The skyrmionium is considered a better information
carrier...

Authors' Reply:

We have changed "*However, the Néel-type skyrmion or skyrmionium is
considered a better information carrier*" to "*The Néel-type skyrmionium is also
considered a better information carrier than skyrmion due to the advantages in
their current-driven dynamics³⁴*" in line 57 (revised version).

3. Line 66: However, the conversion between skyrmions and skyrmioniums is
more complicated and challenging, involving multiple conversions of different
magnetic structures...The experimental work described here does indeed
describe conversions involving several steps, but since some theoretical works
in the bibliography describe single-step conversions, maybe the authors should
point out this difference.

Authors' Reply:

We have changed "*However, the conversion between skyrmions and
skyrmioniums is more complicated and challenging, involving multiple*

conversions of different magnetic textures...” to “The conversion between skyrmions and skyrmioniums is a more complicated and challenging process, involving multiple conversions of different magnetic structures. Although some theoretical reports have described single-step conversions^{25,27,29,30,34}, the experimental realization of reversible conversion between these two topological spin textures requires precise control and manipulation of the domain wall through applied magnetic fields or electric currents.” In line 64 of the revised version.

4. Line 321: In j-l, the field is decreased from 3 mT to 1 mT prevent skyrmion from being destroyed. In j-l, the field is decreased from 3 mT to 1 mT, to prevent the skyrmion from being destroyed.

Authors' Reply:

Thanks for the comment. We have changed “*In j-l, the field is decreased from 3 mT to 1 mT prevent skyrmion from being destroyed.*” to “*In n-q, the field is decreased from 3 mT to 1 mT, to prevent the skyrmion from being destroyed.*” in line 334 of the revised version.

Clarity and context

The procedures and results obtained are well described.

References

I think the relevant literature is reviewed properly.

Providing constructive feedback

I have made my suggestions for the authors in the “Suggested improvements” section.

Reviewers' Comments:

Reviewer #1:

Remarks to the Author:

Yang et al have answered all the comments/suggestions and accordingly revised the manuscript. The revised paper is more understandable, and the results are more solid. I can now recommend it for publication.

Reviewer #2:

Remarks to the Author:

The authors describe the experimental creation of skyrmions and skyrmioniums on a film of Ta(3)/[Pt(0.5)/Co(0.5)]₃/Ru(2)/[Co(0.5)/Pt(0.5)]₃/Ta(3) multilayers. Skyrmions are created, they form clusters that can be turned into skyrmioniums, and turned again into skyrmions, through the action of voltage pulses. The results are relevant for the study of the physical properties of skyrmions and skyrmioniums and to the development of their practical applications.

Significance

The possibility of producing skyrmions and skyrmioniums in a reproducible way at room temperature is very important, both for the investigation of the properties of these systems, and for the development of spintronic applications.

Data and methodology

The presentation is in general clear, and the experimental techniques used are appropriate to obtain the desired nanoscopic structures.

Analytical approach

The analysis of the experiments described in the paper appears to be robust.

Suggested corrections for the manuscript in its present form:

In the revised version of Supporting Material, the following corrections should be made:

Line 145: in the graph, under the horizontal axis it is written HPerpendicular(mT). Instead, it should read BPerpendicular(mT) or $\mu_0 H_{\text{Perpendicular}}$ (mT)

Line 161: instead of HPerpendicular(mT)., it should read BPerpendicular(mT) or $\mu_0 H_{\text{Perpendicular}}$ (mT)

List of Changes

1. The abstract has been refined accordingly.
2. Minor revisions have been made in accordance with the Author Checklist.
3. The main manuscript now includes sections on Data Availability and Code Availability.
4. In response to the comments from Reviewer 2, we have replaced " H_{\perp} " with " $\mu_0 H_{\perp}$ " (unit in mT) in the supplementary information.

Response to Reviewer 2's Comments

Reviewer #2

The authors describe the experimental creation of skyrmions and skyrmioniums on a film of Ta(3)/[Pt(0.5)/Co(0.5)]₃/Ru(2)/[Co(0.5)/Pt(0.5)]₃/Ta(3) multilayers. Skyrmions are created, they form clusters that can be turned into skyrmioniums, and turned again into skyrmions, through the action of voltage pulses. The results are relevant for the study of the physical properties of skyrmions and skyrmioniums and to the development of their practical applications.

Significance

The possibility of producing skyrmions and skyrmioniums in a reproducible way at room temperature is very important, both for the investigation of the properties of these systems, and for the development of spintronic applications.

Data and methodology

The presentation is in general clear, and the experimental techniques used are appropriate to obtain the desired nanoscopic structures.

Analytical approach

The analysis of the experiments described in the paper appears to be robust.

Suggested corrections for the manuscript in its present form:

In the revised version of Supporting Material, the following corrections should be made:

1. Line 145: in the graph, under the horizontal axis it is written HPerpendicular(mT). Instead, it should read BPerpendicular(mT) or μ_0 HPerpendicular(mT)

2. Line 161: instead of HPerpendicular(mT)., it should read BPerpendicular(mT) or μ_0 HPerpendicular(mT)

Authors' Reply:

Thanks for the comments. We have replaced " H_{\perp} " with " $\mu_0 H_{\perp}$ " (unit in mT).